# PARAMETER-FREE MOLECULAR CLASSIFICATION AND REGRESSION WITH GZIP

## ABSTRACT

In recent years, natural language processing approaches to machine learning, most prominently deep neural network-based transformers, have been extensively applied to molecular classification and regression tasks, including the prediction of pharmacokinetic and quantum-chemical properties. However, models based on deep neural networks generally require extensive training, large training data sets, and resource-consuming hyperparameter tuning. Recently, a low-resource and universal alternative to deep learning approaches based on Gzip compression for text classification has been proposed, which reportedly performs surprisingly well compared to large language models such as BERT, given its conceptually simplistic nature. Here, we adapt the proposed method to support multiprocessing, multi-class classification, class-weighing, regression, and multiple modalities and apply it to classification and regression tasks on various data sets of molecules from the organic chemistry, biochemistry, drug discovery, and material science domains. We further propose converting numerical descriptors into string representations, enabling the integration of language input with domain-informed descriptors. Our results show that the method can be used to classify and predict a variety of properties of molecules or the binding affinity of protein-ligand complexes, can reach the performance of transformers and graph transformers in a subset of tasks, and has the potential for application in information retrieval from large chemical databases.

## 1 INTRODUCTION

Machine learning methods to classify or predict the properties of molecules have become omnipresent tools in chemical and biological research. Classification tasks include categorising molecules into toxic and non-toxic, protein-binding and non-binding, or otherwise pharmacological active or inactive compounds. Meanwhile, regression tasks encompass predicting various physicochemical and pharmacological properties, such as solubility and lipophilicity, protein-ligand binding affinity, or even quantum chemical properties. With the rise of deep learning during the past decade, molecular classification and property prediction have increasingly been carried out by ever-larger models with mixed results, as in tasks such as pharmacokinetic property prediction, where data remains scarce, deep learning methods have yet to perform significantly better than ensemble methods (Muratov et al., 2020). Across all machine learning approaches, the most utilised methods are fingerprint-, SMILES-, and graph-based approaches, where molecular feature vectors, text representations of molecules, and molecular graphs, respectively, are used as the input of the respective class of models (MLPs, transformers, and GNNs) (Probst & Reymond, 2018; Reiser et al., 2022; Ross et al., 2022). Even though the text-based SMILES encoding of a molecule is often called a 1D representation (as opposed to the "2D" molecular graph and the 3D molecular structure), a SMILES string contains all information of its respective molecular graph, as it is constructed by traversing said graph using a depth-first search (DFS) algorithm (Weininger, 1988). Furthermore, it also contains implicit and explicit information on the 3D structure of the molecule, as molecular structure is tied to molecular topology, and molecular chirality is often directly defined using the SMILES notation.

Recently, a parameter-free text classification approach based on Gzip compression has been proposed, which has shown excellent performance compared to deep learning architectures, such as transformers, on text-classification benchmark data sets (Jiang et al., 2022). The intuition guiding

the method is to exploit the capability of lossless compressors, such as Gzip, to capture regularity using a statistical model that enables to assign shorter codes to high-probability sequences. It is then assumed that texts in the same category share similar regularity and are thus close in compression space under a normalised compression distance (NCD) metric (Li et al., 2004). A k-nearest neighbour classifier is then used to classify text under the NCD metric. As the SMILES string encoding of molecular graphs has proved to be a well-performing molecular representation for applying other NLP methods, such as transformers or locality-sensitive hashing (LSH) (Schwaller et al., 2019; Probst & Reymond, 2018), we hypothesise that the methodology presented by Jiang et al. (2022) will also yield good results for chemical tasks, and a comparison of the other NLP-based and -inspired methods with compression-based representation learning is warranted.

Here, we report an implementation of the Gzip-based text representation method, initially introduced by Jiang et al. (2022), targeted towards chemical machine learning problems. We present two algorithms denoted MolZip and MolZip-Vec, both capable of single- and multimodal molecular classification and regression, with MolZip-Vec also allowing for the incorporation of real-valued vectors to embed precomputed chemical values. We compare our implementation to deep neural network-based methods, such as transformers and graph neural networks (GNNs), on molecular classification and regression tasks that include a multimodal binding-affinity prediction problem. We show that this conceptually simple and inexpensive method works not only for the classification and clustering of data in a natural language processing context but also on SMILES-encoded molecules without requiring time-consuming training on specialised hardware, such as GPUs. More than that, we extend the methodology to support most chemical machine-learning tasks through an open-source Python library.

## 2 Results & Discussion

We benchmark the proposed methodology using the MoleculeNet benchmark for molecular machine learning and compare it against ChemBERTa-1, ChemBERTa-2, and GROVER$_{large}$ (Wu et al., 2018; Yang et al., 2019; Rong et al., 2020; Chithrananda et al., 2020; Ahmad et al., 2022). ChemBERTa-1 and ChemBERTa-2 are BERT-based transformers pre-trained on 10 million molecules and then fine-tuned on the specific tasks from the benchmark. ChemBERTa-2 comes in a masked language model (MLM) and a multi-task regression (MTR) variant; the latter is pretrained targeting the same 200 RDKit-computed properties we include in MolZip-Vec. GROVER$_{large}$ is a graph transformer based on graph attention networks (GAT) first introduced by Veličković et al. (2018) that was trained on 10 million molecules. We chose these three architectures as they represent basic implementations of the two most common types (SMILES and graph-based) of transformers used in cheminformatics. A recent effort, Molformer-XL Ross et al. (2022), which was trained on 1.1 billion molecules for approximately 208 hours on 16 NVIDIA V100 GPUs and then fine-tuned for another 12 hours, has not been included in Tables 1 and 2 in order to compare architectures that represent early efforts on the respective methodologies. Furthermore, we extended MolZip towards predicting protein-ligand binding affinities and compared the approach to graph neural network-based methods, which have seen continuous use and advancements over the past years.

### 2.1 Classification

We follow the proposed method by Jiang et al. (2022) for the classification tasks and extend it with multiprocessing and nearest-neighbour weighing to support imbalanced data sets better. In addition, we implement a framework which provides serialisable text transformations on the input SMILES, including the translation into alternative string-based molecular representations (DeepSMILES and SELFIES) and SMILES-based augmentation, which augments a sample by concatenates a user-chosen number of different valid SMILES representations of a given molecule (Bjerrum, 2017; O'Boyle & Dalke, 2018; Krenn et al., 2022). For both MolZip and MolZip-Vec, we choose the parameter $k = 5$ for the k-nearest neighbour classification and assume that all data sets are imbalanced, therefore adjusting the kNN classification based on class weights that are calculated using the scikit-learn (v1.3.1) utility function `compute_class_weight`.

Before benchmarking and comparing transformer-based methods, we evaluated the effect of translation and augmentation transformations. Table A.1 compares the performance of SMILES, DeepSMILES, and SELFIES-encoded molecules with otherwise default parameters ($k = 5$, no

augmentation) on various data sets. Based on these results, we decided to use SMILES encoding for our implementation, as it provides a balanced baseline across all evaluated data sets. However, the result of the SELFIES-encoding on the BACE (regression) data shows that the encoding could indeed have a strong influence on the observed performance, as the SMILES and DeepSMILES-encodings reached a performance of 0.668 and 0.682, respectively, under the AUROC metric, the SELFIES-encoded variant reached 0.720. However, this observation was an outlier rather than a trend. Evaluating the effect of augmentation, which concatenates multiple variants of SMILES-encodings of the same molecule (e.g. starting the depth-first search, which constructs the SMILES, at a different atom), using the BBBP and BACE (classification) sets showed mixed results. While the performance of MolZip on the BACE (classification) data set could have been pushed by approximately 10% (Figure A.1b), a lack of correlation of the positive effect on the validation and test set, as well as generally lower performance on the BBBP set (Figure A.1a), led us to report the non-augmented classification metrics. The same holds for MolZip-Vec, as presented in Figure A.1.

Table 1: Performance comparison between ChemBERTa-1, ChemBERTa-2, GROVER, MoZip, and MolZip-Vec. Underlined results are the best between ChemBERTa-1 and MolZip, bold results are the best overall. Note: The data for GROVER$_{large}$ is taken from Zhou et al. (2023), as the authors of the original GROVER paper did not include benchmark results for scaffold splits.

| Data Set | Split | Metric | ChemBERTa-1 | MolZip (ours) | ChemBERTa-2 (MLM) | ChemBERTa-2 (MTR) | MolZip-Vec (ours) | GROVER (large) |
|---|---|---|---|---|---|---|---|---|
| BBBP | scaffold | AUROC | 0.643 | 0.665 | 0.696 | **0.733** | 0.692 | 0.695 |
| ClinTox (CT_TOX) | scaffold | AUROC | 0.733 | **0.862** | 0.349 | 0.601 | 0.500 | - |
| Tox21 (SR-p53) | scaffold | AUROC | 0.728 | 0.692 | 0.748 | **0.827** | 0.663 | - |
| HIV | scaffold | AUROC | 0.622 | **0.684** | - | - | **0.684** | 0.682 |
| BACE (classification) | scaffold | AUROC | - | 0.667 | 0.729 | 0.783 | 0.668 | - |

The results reported in Table 1 show that our compression-based methods reach competitive performance compared to baseline implementations of transformers. On a head-to-head with the original seq2seq implementation of ChemBERTa-1, MolZip performs better on 3 out of 4 data sets. However, it falls short compared to the second-generation ChemBERTa-2, which was trained using a masked language modelling (MLM) approach and a multi-task regression (MTR) pretraining scheme, targeting 200 precomputed molecular descriptors provided by RDKit. The same 200 RDKit descriptors were embedded with their respective SMILES representation of the molecule for MolZip-Vec, which fails to improve on MolZip on the classification tasks. Furthermore, MolZip and MolZip-Vec perform on par with the graph transformer GROVER$_{large}$ on BBBP and HIV. These results indicate that, like with natural language, the compression-based method is surprisingly performant compared to relatively large models such as a BERT transformer or a GAT-based graph transformer.

## 2.2 REGRESSION

We implemented regression functionality by taking the arithmetic mean of the k-nearest neighbours weighted by the inverse of their normalised compression distance NCD to the query (Equation 3). For all regression tasks, we choose $k = 25$ to potentially smooth noise labels. The two physical chemistry regression tasks (Delaney/ESOL and Lipophilicity [LIPO]), for which data was available for ChemBERTa-2 and GROVER, and the two pharmacokinetics regression tasks (BACE [regression] and Clearance), for which data was only available for ChemBERTa-2, were used to benchmark our regression implementation. As for the classification tasks, we evaluated the SMILES-encoding against DeepSMILES and SELFIES and again chose SMILES over the two alternatives for benchmarking (Table A.1). We further evaluated the effects of augmentation for regression tasks on the two data sets Delaney/ESOL and BACE (regression). Interestingly, and unlike our evaluation of augmentation on classification tasks, augmentation on regression tasks has a general, and in some cases significant, positive effect on performance (Figure A.1c,d): Augmenting each SMILES in the Delaney/ESOL data set with an additional 19 SMILES, that represent the same molecule but differ in atom-order, would decrease the RMSE measured for MolZip by 28% from 1.510 to 1.097. However, we omit reporting augmentation-based results for the regression tasks to be compatible with the results reported for the classification tasks.

The results reported in Table 2 show that our compression-based methods reach competitive performance compared to ChemBERTa-2 in two out of four tasks; however, they perform comparatively poor on the two data sets for which metrics were available for GROVER. For ChemBERTa-1, no re-

Table 2: Performance comparison between ChemBERTa-1, ChemBERTa-2, GROVER, MoZip, and MolZip-Vec. Underlined results are the best between ChemBERTa-1 and MolZip, bold results are the best overall. Note: The data for GROVER (large) is taken from Zhou et al. (2023), as the authors of the original GROVER paper did not include benchmark results for scaffold splits.

| Data Set | Split | Metric | ChemBERTa-1 | MolZip (ours) | ChemBERTa-2 (MLM) | ChemBERTa-2 (MTR) | MolZip-Vec (ours) | GROVER (large) |
|---|---|---|---|---|---|---|---|---|
| Delaney/ESOL | scaffold | RMSE | - | 1.510 | 0.961 | **0.858** | 1.271 | 0.895 |
| BACE (regression) | scaffold | RMSE | - | 1.174 | 1.611 | 1.417 | **1.133** | - |
| LIPO | scaffold | RMSE | - | 1.042 | 1.009 | **0.744** | 0.915 | 0.823 |
| Clearance | scaffold | RMSE | - | 49.885 | 53.859 | **48.93** | 49.211 | - |

gression benchmarks were available. Unlike in the classification benchmark, MolZip-Vec performed better than MolZip on all data sets. Nevertheless, while it performed better than ChemBERTa-2 (MLM) on 3 out of 4 data sets, it was only able to perform better than ChemBERTa-2 (MTR) on BACE (regression). These results show that compression-based classification on molecules can be successfully extended to regression.

## 2.3 BINDING AFFINITY PREDICTION

In addition to molecular property prediction, we tested the ability of the compression-based approach to predict protein-ligand binding affinities—an essential metric for rational drug design, which aims to find a drug candidate, given structural information on a disease-associated protein (Gane & Dean, 2000). The protein-ligand binding affinity describes whether and how strong a ligand binds non-covalently to a protein, usually causing a conformational change of the protein and potentially leading to a therapeutic effect (Williams, 2013). The prediction of the binding affinity, given a potential ligand and a protein's structure or amino acid sequence, is therefore of interest to computational chemistry. Over the past years, geometric deep learning, specifically graph neural network-based approaches, have emerged as the most investigated methods to predict binding affinities, as they are capable of capturing topological and spatial features important to protein-ligand binding (Li et al., 2021; Méndez-Lucio et al., 2021; Nguyen et al., 2021).

Table 3: Performance of MolZip and MolZip-Vec on the PDBbind data set compared to graph representation learning-based methods. [*]Augmented with an additional SMILES.

| Model | | RMSE | MAE | R |
|---|---|---|---|---|
| GraphDTA Methods | GCN | 1.735±0.034 | 1.343±0.037 | 0.613±0.016 |
| | GAT | 1.765±0.026 | 1.354±0.033 | 0.601±0.016 |
| | GIN | 1.640±0.044 | 1.261±0.044 | 0.667±0.018 |
| | GAT-GCN | 1.562±0.022 | 1.191±0.016 | 0.697±0.008 |
| GNN-based Methods | SGCN | 1.583±0.033 | 1.250±0.036 | 0.686±0.015 |
| | GNN-DTI | 1.492±0.025 | 1.192±0.032 | 0.736±0.021 |
| | D-MPNN | 1.493±0.016 | 1.188±0.009 | 0.729±0.006 |
| | MAT | 1.457±0.037 | 1.154±0.037 | 0.747±0.013 |
| | DimeNet | 1.453±0.027 | 1.138±0.026 | 0.752±0.010 |
| | CMPNN | **1.408±0.028** | **1.117±0.031** | **0.765±0.009** |
| Compression-based Methods | MolZip[*] (ours) | 1.447±0.031 | 1.134±0.020 | 0.746±0.013 |
| | MolZip-Vec (ours) | 1.675±0.000 | 1.300±0.000 | 0.648±0.000 |

To tackle the challenge of protein-ligand binding affinity prediction using MolZip and MolZip-Vec, we implemented a data loader capable of loading and concatenating different modalities, namely SMILES and amino acid sequences, and pass it to a MolZip or MolZip-Vec regressor. As we evaluated the method on the PCBbind data set (Liu et al., 2015), the following information was provided for each protein-ligand complex: (i) structural and compositional data for the ligand, (ii) structural and compositional data for amino acids that are part of the binding pocket of the protein, and (iii) structural and compositional data for the entire protein. From this data, we generated the following encodings: (1) For the ligand, a SMILES string, (2) for the binding pocket, a SMILES string and

a one-letter amino acid string sequence, where amino acids that are not part of the binding pocket are replaced by an *X*, and (3) for the protein, a one-letter amino acid string sequence. These encodings provided us with four modalities (one molecule representation, two binding pocket representations, and one whole-protein representation) that can be combined arbitrarily through concatenation. Exploratory benchmark results for the combinations ligand (SMILES), binding pocket (SMILES), binding pocket (amino acid sequence), whole-protein (amino acid sequence), ligand (SMILES) + binding pocket (SMILES), ligand (SMILES) + binding pocket (amino acid sequence), and ligand (SMLIES) + whole-protein (amino acid sequence) can be found in Table A.2. The combination ligand (SMILES) + binding pocket (amino acid sequence) provided the best results.

Comparing our best results against baseline GraphDTA- and GNN-based methods, it becomes evident that MolZip performs exceptionally well. It does not only perform better than basic GNNs, including GCN, GAT, and GIN, that used atom features as node attributes for the molecular graph and the protein sequence as inputs (Nguyen et al., 2021), but also better than methods that include geometric information in the form of atom-wise protein-ligand interactions, such as GNN-DTI (Lim et al., 2019). Introducing molecular descriptors with MolZip-Vec reduces the performance to that of GraphDTA methods, hinting at the importance of a relatively fuzzy representation of the ligand to a well-performing compression-based model.

## 2.4 IMPLICATIONS FOR CHEMICAL INFORMATION RETRIEVAL

Compression-based representation of molecules may have implications beyond machine learning. As chemical databases such as ZINC or GDB contain billions of molecules, and even partially human-curated databases like PubChem contain more than 100 million unique molecules, retrieving information based on chemical features is growing increasingly important (Irwin & Shoichet, 2005; Visini et al., 2017; Kim et al., 2019). Currently, most searches rely on graph topological similarity based on molecular fingerprints, precomputed stored chemical descriptors, or a combination of both (Warr et al., 2022). With the findings presented in this study, we have shown that the lossless compression-based combination of molecular structure and chemical descriptors, used as an input for MolZip-Vec, presents a low-memory alternative to established methods discussed by Warr et al. (2022) that allows for direct structure and property-based storage, similarity search and indexing.

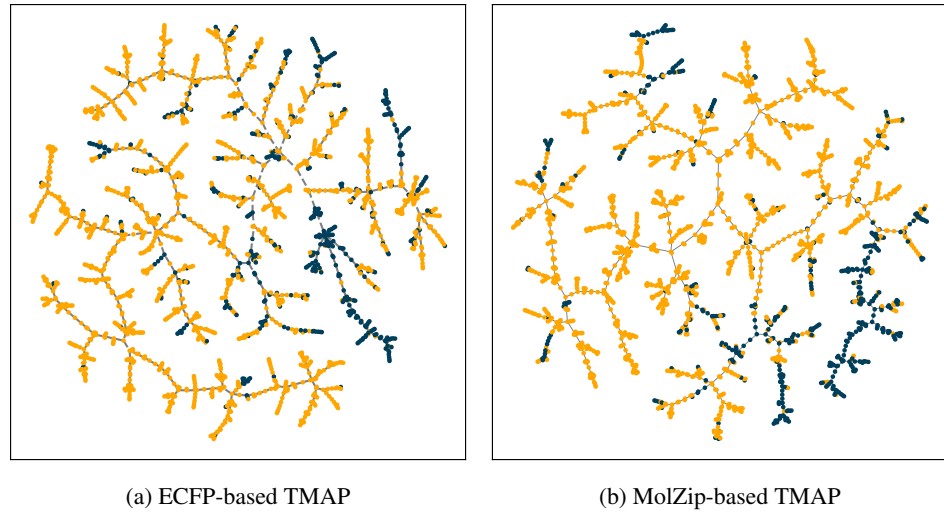

(a) ECFP-based TMAP       (b) MolZip-based TMAP

Figure 1: TMAP visualisation of the BBBP data set.

The ability to index and search molecules similar to a commonly used molecular fingerprint, ECFP (extended-connectivity fingerprint) (Rogers & Hahn, 2010), is apparent when visually inspecting the TMAP plots in Figure 1, where the high-dimensional ECFP and compression spaces of the BBBP data set are visualized by embedding a minimum spanning tree calculated in the original spaces in the Euclidean plane. They both show similar clusters of molecules capable of passing the blood-brain barrier.

## 3 METHODS

### 3.1 IMPLEMENTATION

We implemented the compression-based classification according to the code presented in the original preprint and extended it with support for multiprocessing, class weights, multi-task classification, and regression (Jiang et al., 2022). As in the original preprint, the Normalised Compression Distance (NCD) between molecules $x$ and $y$ is calculated as

$$NCD(x, y) = \frac{C(xy) - \min\{C(x), C(y)\}}{\max\{C(x), C(y)\}} \tag{1}$$

where $C(x)$ and $C(y)$ are the compressed lengths of the SMILES representations of molecules $x$ and $y$, respectively, and $C(xy)$ the compressed length of the concatenated SMILES representations of the two molecules. We changed the implementation of the k nearest-neighbour classifier by weighting the class counts $C_i$ among the k nearest-neighbours using the formula

$$Cw_i = C_i W_i (1 - \bar{d}_i) \tag{2}$$

where $Cw_i$ are the weighted class counts among the k nearest neighbors, $W_i$ the class weights computed from class distribution in the training data set and $\bar{d}_i$ the mean distance (NCD) between the query point and the k nearest neighbours of class $i$. The class weights were computed using the function `compute_class_weight` from the Python package scikit-learn. For the k nearest-neighbour regression, a simple distance weighted kNN regressor was implemented in the form of

$$y_i = \frac{\sum_j^k y_j (1 - \bar{d}_{ij})}{\sum_j^k (1 - \bar{d}_{ij})} \tag{3}$$

where $\bar{d}_{ij}$ is the distance (NCD) between the query point $i$ and the k nearest neighbours $j$, $y_j$ the values of the k nearest neighbours, and $y_i$ the predicted value.

Multiprocessing has been implemented using the Python standard library (`multiprocessing`).

### 3.2 MOLZIP-VEC

For MolZip-Vec, we combined SMILES strings with numerical descriptors of molecules commonly used in chemoinformatics. Specifically, we utilized a vector comprising 200 molecular descriptors from the RDKit cheminformatics library RDKit (2023), which are typically used to augment graphs in molecular graph representation learning (Yang et al., 2019). A complete list of the 200 descriptors can be found in the documentation of the descriptastorus (v2.6.1) Python package. In order to combine and compress the numerical descriptors with the molecular string representation, the values are binned and subsequently translated into a set of non-ASCII Unicode characters. The three molecular string representations (SMILES, DeepSMILES, and SELFIES) used in this work only use ASCII characters, so collisions are avoided. Empirically, we found that 256 is a suitable number of bins. A special character prefixes negative values to represent positive and negative bins distinctly. Each string-based representation of the numerical vector is concatenated to the corresponding SMILES string, significantly improving the RMSE of several datasets listed in Table 2. In Figure A.2, we show that including numerical vectors increasingly improves the performance with growing training set size on the FreeSolv data set (Mobley, 2013). Note that the computational cost for the prediction is slightly higher because of the increased string length.

### 3.3 BENCHMARKING

The benchmarking results and details of ChemBERTa-1, ChemBERTa-2, and MOLFORMER-XL were taken from the respective publications (Chithrananda et al., 2020; Ahmad et al., 2022; Ross et al., 2022). For GROVER, the benchmark results based on scaffold splits have been taken

from Zhou et al. (2023). Benchmark results for GraphDTA and GNN-based methods were taken from Li et al. (2021).

All benchmarks were run on a Intel Core i7-13700K CPU with a total of 16 cores (8 performance and 8 efficiency cores) with a maximum power-draw of 253W. Together, all classification and regression benchmarks took 43h 55m to complete. All energy came from renewable sources (hydropower and solar energy).

## 4 CONCLUSION

By applying the proposed Gzip-based text classification method by Jiang et al. (2022) to multiple molecular classification tasks and extending it to regression problems, we verified its validity and utility beyond natural language processing tasks. While the proposed method does not achieve the state-of-the-art performance set by the latest iteration of transformers trained on billions of molecules on benchmark tasks, it performs as well as baseline BERT- and GAT-based transformers. Furthermore, it is highly intriguing that a method based on differences in the length of Gzip compressed string representations of molecules can yield comparable or even superior performance compared to deep learning models. We have also shown that the methodology can be extended to multimodal binding affinity tasks, where SMILES strings and amino acid sequences are jointly compressed. On the PDBbind data set, our proposed method performs better than all GraphDTA- and most GNN-based methods, including those incorporating spatial information. Additionally, we have demonstrated that integrating molecular SMILES strings with string-converted chemical descriptors can significantly enhance the accuracy compared to using SMILES input alone. Finally, we discuss how such a method could be of interest outside machine learning and support a new generation of chemical information retrieval in ultra-large databases. However, certain limitations and challenges still need to be addressed, including the relatively high time complexity of the kNN-based approach and the elucidation of the reasons for significant gaps in performance on specific data sets compared to the state-of-the-art.

## ACKNOWLEDGMENTS

*Anonymized*

## CODE AND DATA AVAILABILITY

All code and data, with exception of the PDBbind data set, is available in the associated anonymized repository `https://anonymous.4open.science/r/molzip-02F0`. The PDBbind data set is available for download here: `http://www.pdbbind.org.cn/`.

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

# A APPENDIX

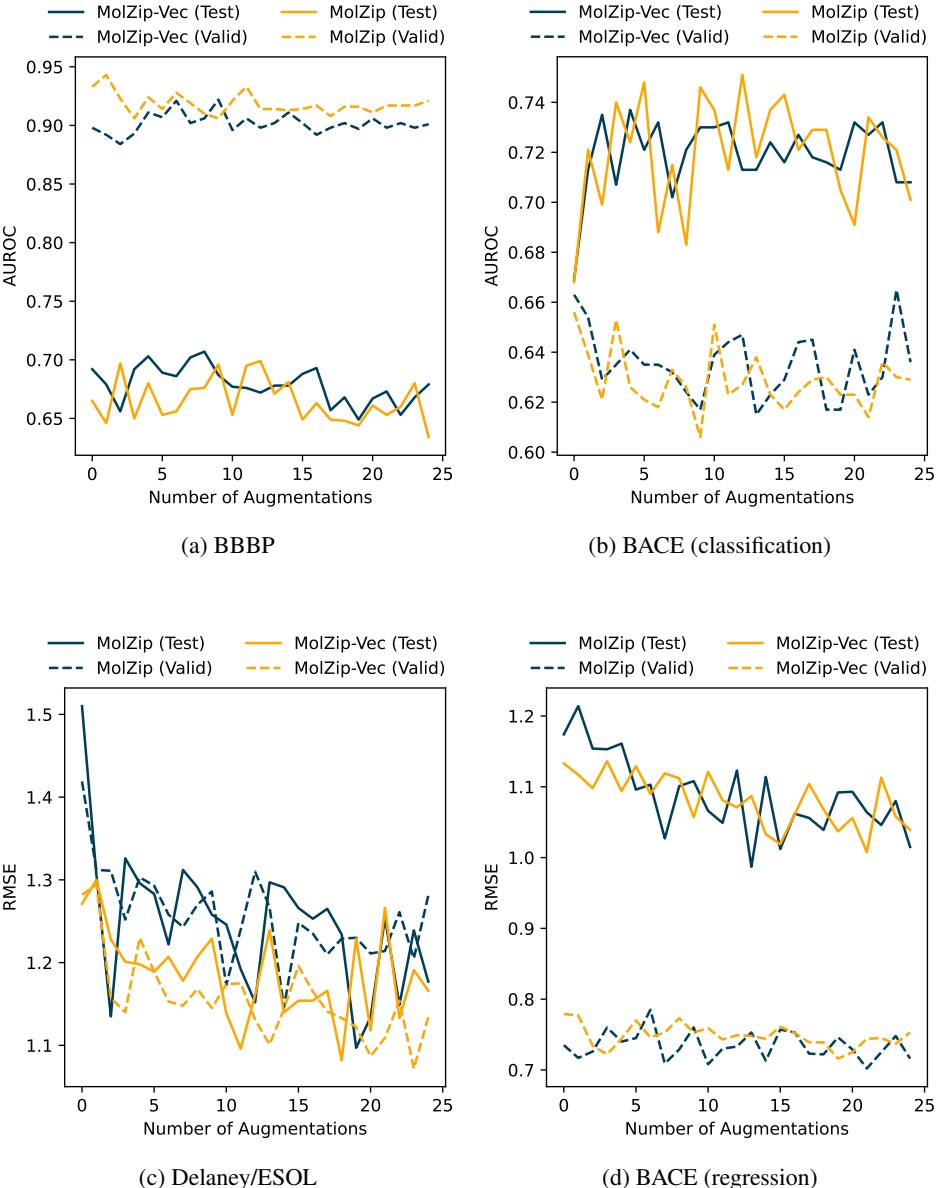

Figure A.1: Influence of data augmentation (randomised SMILES) on validation and test results.

Table A.1: Effect of different string-encodings of molecules on MolZip performance.

| Data Set | Split | Metric | SMILES | DeepSMILES | SELFIES |
|---|---|---|---|---|---|
| BBBP | scaffold | AUROC | 0.665 | **0.688** (+0.023) | 0.648 (-0.017) |
| ClinTox (CT_TOX) | scaffold | AUROC | 0.862 | **0.898** (+0.036) | 0.723 (-0.139) |
| Tox21 (SR-p53) | scaffold | AUROC | 0.692 | **0.694** (+0.002) | 0.681 (-0.011) |
| HIV | scaffold | AUROC | **0.684** | 0.660 (-0.024) | 0.660(-0.024) |
| BACE (classification) | scaffold | AUROC | 0.668 | 0.682 (+0.014) | **0.720** (+0.052) |
| Delaney/ESOL | scaffold | RMSE | **1.510** | 1.519 (+0.009) | 1.738 (+0.228) |
| BACE (regression) | scaffold | RMSE | 1.174 | 1.211 (+0.035) | **1.140** (-0.034) |
| LIPO | scaffold | RMSE | **1.042** | 1.045 (+0.003) | 1.069 (+0.027) |
| Clearance | scaffold | RMSE | **49.885** | 51.116 (+1.231) | 50.540 (+0.655) |

Table A.2: Effect of different combinations of PDBbind modalities on the performance of MolZip (without augmentations).

| Modalities | RMSE | MAE | R |
|---|---|---|---|
| Ligand (SMILES) | 1.776 | 1.416 | 0.591 |
| Pocket (SMILES) | 1.653 | 1.311 | 0.653 |
| Pocket (AA Seq.) | 1.665 | 1.303 | 0.644 |
| Protein (AA Seq.) | 1.885 | 1.512 | 0.525 |
| Ligand (SMILES) + Pocket (SMILES) | 1.598 | 1.258 | 0.679 |
| Ligand (SMILES) + Pocket (AA Seq.) | **1.504** | **1.187** | **0.721** |
| Ligand (SMILES) + Protein (AA Seq.) | 1.688 | 1.307 | 0.633 |

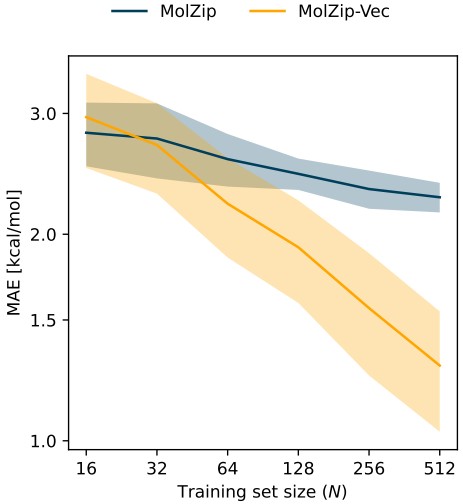

Figure A.2: Comparing SMILES and a combination with molecular property vectors SMILES & Vector). Learning curves *i.e.* mean absolute error (MAE) evaluated using 10-fold random splits of the FreeSolv(Mobley, 2013) database for solvation free energies. The x-axis shows the number of training examples $N$ added at constant test set size. The curves show the average over the splits and the shadow the standard deviation.

