# OpenReview forum: "Parameter-Free Molecular Classification and Regression with Gzip"
_ICLR.cc/2024/Conference — Submitted to ICLR 2024_

### Official Review · Reviewer_ovMF · 2023-10-18

**Soundness:** 3 good
**Presentation:** 3 good
**Contribution:** 2 fair
**Rating:** 5
**Confidence:** 3

**Summary:**

In "Parameter-free Molecular Classification and Regression with Gzip", the authors investigate the applicability of a recently proposed text representation method based on Gzip compression (by Jiang et al.) for cheminformatics applications. The core concept involves comparing the compressed lengths of concatenations of pairs of strings, with the assumption that concatenations of similar strings are more compressible than those of dissimilar strings. This enables the definition of a "distance" between strings, which can then be utilized in standard learning algorithms like $k$-nearest neighbors. The authors of the present paper apply this idea to text-based molecular representations (e.g. SMILES strings) to perform various classification and regression tasks, e.g. predicting binding affinities or toxicity. They compare two Gzip-based methods (MolZip and MolZip-Vec) to various deep neural network-based methods on a range of benchmarks. Their findings demonstrate that the parameter-free Gzip-based methods achieve competetive results, sometimes even surpassing the more complex deep learning-based approaches.

**Strengths:**

The paper is a straightforward application of a recently proposed text representation method to cheminformatics problems. The underlying concepts are well-described and the text is clearly written. An aspect of the paper which I find particularly commendable is the authors' transparency when it comes to the reported results. For example, even though they could get improved results for regressions tasks with slight modifications to their method (augmenting the text descriptor with additional SMILES strings representing the same molecule), they still chose to report "baseline numbers" in tables with performance comparisons, so that results between regression and classification tasks are compatible.

**Weaknesses:**

Although the authors go great lengths to compare compression-based methods to a variety of established deep learning-based methods on several benchmarks, in my opinion, their results raise an important question, which the current experiments do not answer: *Why* do the compression-based methods perform so well? Is it because compression-based text representations for molecules are surprisingly powerful, or is it because the tasks themselves are actually very simple to solve, i.e., deep learning-based methods are simply overengineered? I believe the paper would be greatly improved if this question was answered conclusively.

**Questions:**

1. The definition of normalised compression distance (NCD) feels unintuitive to me. For reference, it is given as:
$$
NCD(x,y) = \frac{C(xy) - \min\\{C(x), C(y)\\}}{\max\\{C(x), C(y)\\}}
$$
where $C(x)$ is the compressed length of the string $x$ and $xy$ denotes the concatenation of strings $x$ and $y$. I calculated the NCD for the SMILES strings
- $x_1$: `'CC(C)CC(C(=O)O)N'` (leucine)
- $x_2$: `'CCC(C)C(C(=O)O)N'` (isoleucine)
- $x_3$: `'C1=CC=C(C=C1)N'` (aniline)
- $x_4$: `'C1=CC=C2C=CC=CC2=C1'` (naphthalene)
and arrived at the distance matrix
$$
D = \begin{bmatrix}
 0.088 & 0.176 & 0.441 & 0.457 \\\\
 0.118 & 0.088 & 0.441 & 0.457 \\\\
 0.382 & 0.382 & 0.094 & 0.286 \\\\
 0.429 & 0.457 & 0.343 & 0.086
\end{bmatrix}
$$
with $D_{ij} = NCD(x_i, x_j)$. While the entries generally make sense to me (e.g. leucine is closer to isoleucine than it is to aniline), it becomes immediately apparent that the NCD is not a proper metric. For example, $NCD(x_i, x_j) \neq NCD(x_j, x_i)$ (not symmetric) and $NCD(x_i, x_i) > 0$ (the distance from a point to itself is not zero). To me, it seems like these properties should negatively impact the performance of algorithms like $k$-nearest neighbors.
A relatively simple alternative definition given by
$$
NCD(x,y) = \frac{0.5[C(xy) + C(yx)] - \min\\{C(xx), C(yy)\\}}{\max\\{C(xx), C(yy)\\}}
$$
instead leads to the distance matrix
$$
D = \begin{bmatrix}
 0.000 & 0.054 & 0.297 & 0.329 \\\\
 0.054 & 0.000 & 0.297 & 0.342 \\\\
 0.297 & 0.297 & 0.000 & 0.211 \\\\
 0.329 & 0.342 & 0.211 & 0.000
\end{bmatrix}
$$
While this is still not a proper metric (nothing guarantees that non-identical points have a distance larger than zero and the triangle inequality probably does not hold), it at least has more of the desired properties. Can the authors comment on the definition used in the paper and why it is used? If possible, I would appreciate some experiments with the more "metric-like" definition. Does it work better in practice? If not, can the authors give reasons why the definition given in the paper works better?
(Note: I understand that the authors took the definition of the NCD from Jiang et al. and did not come up with it themselves, but I hope they can still answer my questions.)

2. I think the paper would benefit from additional "simple baselines", so that it becomes clearer whether compression-based methods are surprisingly good, or the investigated problems are just surprisingly simple to solve (see Weaknesses). I could imagine a descriptor inspired by Bag of Bonds (see 10.1021/acs.jpclett.5b00831) as a simple baseline: For example, a water molecule might be described by the vector `[2 0 1 0 2 0]` and a methane molecule by `[4 1 0 4 0 0]`, where the entries `[#H #C #O #CH #OH #CO]` correspond to atom (e.g. `#H`: number of hydrogen atoms) and bond (e.g. `#CH`: number of C-H bonds) counts (this descriptor can be trivially extended to more atom/bond types). As a "distance metric", I suspect cosine similarity might work well for this kind of descriptor (probably better than Euclidean distance).

3. Related to the point above: For MolZip-Vec, a vector of 200 molecular descriptors is converted to a string (by binning numerical values and translating them to non-ASCII characters), and appended to the SMILES string. This seems to be particularly effective for regression tasks. How does a simple baseline, e.g. linear regression on the raw vector of molecular descriptors, perform? Alternatively, have the authors considered using e.g. the cosine similarity between the raw vectors of molecular descriptors as a distance metric (instead of the NCD)? What's the performance?

---

### Official Review · Reviewer_5hym · 2023-11-03

**Soundness:** 1 poor
**Presentation:** 1 poor
**Contribution:** 2 fair
**Rating:** 3
**Confidence:** 4

**Summary:**

This paper applies the gzip classification method from Jiang et al 2022 to chemical datasets, showing experimentally that the performance on some MoleculeNet classification and regression datasets are similar to fine-tuned versions of large chemical language models.

**Strengths:**

The main strength of the paper is that it runs an interesting experiment: how well does gzip classification/regression perform on molecule datasets? This is an interesting number to add to the set of established baselines in molecular property prediction.

**Weaknesses:**

I viewed the following aspects of the paper as weak, which have led me to recommend rejection:

- Magnitude of contribution is rather small: while this is a nice experiment to run, ultimately the paper asks and answers a pretty simple question: what is the performance of the gzip method? MoleculeNet is small and can be easily obtained online. Similarly, the gzip method is quite simple to code. This experiment seems like < 1 week of work. To me, these results are more suitable for a blog post or a workshop than as a paper in a top-tier ML conference.
- Poor presentation: I felt that the paper was not well-written. To me, the sensible structure for such a paper would be to describe the method of Jiang et al 2022, describe how it was modified for molecules, present experimental results, then discuss impact. Instead, the paper presents experimental results, then only describes how the method differs from Jiang et al 2022 without describing that method. I think the paper is not readable as a standalone document in its current form: the reader would also need to read Jiang et al 2022 in order to understand this paper.
- Doesn't consider classical baselines: this paper seems to generally mirror the narrative of Jiang et al 2022 which compares gzip against large fine-tuned language models. While such a narrative seems appropriate for NLP, machine learning for molecules is dominated by small datasets where the performance of classical methods is competitive (e.g. SVM or KNN on molecular fingerprints). I think the paper should discuss this and include results from classical baselines. My guess is that the gzip KNN's performance is not too different to the performance of KNN on fingerprints (using an appropriate distance metric for fingerprints such as Jaccard distance).
- Paper does not contain error bars on any experiments, despite all methods clearly containing randomness. Therefore the significance of performance differences is unclear. In my experience working with these datasets the standard deviations were quite high.

**Questions:**

- Can you provide performance of classical ML methods on molecular fingerprints as baselines?
- What are the error bars / measure of statistical variation for the experiments performed
- In my review above, I basically stated that the contribution is "not enough" for ICLR. Do you have a counterargument for the significance of your contribution?

---

### Official Review · Reviewer_1REg · 2023-11-06

**Soundness:** 3 good
**Presentation:** 3 good
**Contribution:** 3 good
**Rating:** 8
**Confidence:** 4

**Summary:**

This paper proposes a method for molecular classification and regression based on Gzip compression of SMILES string representations of molecules. The main contributions are:

* Compression-based representation learning: The paper adapts the method of Jiang et al. (2022) to use the normalized compression distance (NCD) of SMILES strings as a similarity metric for molecules. The paper also extends the method to support regression, multimodal input, and numerical descriptors.
* Benchmarking on molecular tasks: The paper evaluates the performance of the compression-based method on various molecular classification and regression tasks from the MoleculeNet benchmark, as well as a binding affinity prediction task using the PDBbind data set. The paper compares the results with transformer-based and graph neural network-based methods.
* Implications for chemical information retrieval: The paper discusses how the compression-based method can be used for indexing and searching molecules in large chemical databases based on structure and property similarity. The paper also provides a visualization of the compression space using TMAP.

**Strengths:**

The paper introduces a new approach to molecular learning that leverages the power of compression algorithms to capture the structural and chemical similarity between molecules. The paper adapts an existing method of similarity measurement based on normalized compression distance (NCD) to the domain of molecular data. It extends it to support regression, multimodal input, and numerical descriptors. The paper also shows how the compression-based method can be used for indexing and searching molecules in large chemical databases based on both structure and property similarity. Overall, the paper presents a fresh perspective on molecular learning that is both intuitive, effective, and parameter-free.

The authors perform a rigorous evaluation of the proposed method on various molecular classification and regression tasks from the MoleculeNet benchmark, as well as a binding affinity prediction task using the PDBbind data set. The paper compares the results with transformer-based and graph neural network-based methods. It shows that the compression-based method achieves comparable or better performance than these methods, especially when using multimodal input. The paper also discusses the limitations and future directions of the proposed method, such as handling rare or unseen molecules, improving scalability and interpretability, and integrating with other machine learning models.

The work has several implications for molecular learning and chemical information retrieval. The compression-based method offers a parameter-free and universal way of representing molecules that can handle various modalities and tasks without requiring training or tuning. The method can be used for indexing and searching molecules in large chemical databases based on both structure and property similarity, which can accelerate drug discovery and material design. The method can also be combined with other machine learning models to improve performance or interpretability. Overall, the paper presents a promising direction for future research in molecular learning.

**Weaknesses:**

The paper builds on an existing method of similarity measurement based on normalized compression distance (NCD) and applies it to the domain of molecular data. While this work extends the technique to support regression, multimodal input, and numerical descriptors, it does not introduce fundamentally new concepts or techniques. The paper also does not compare the proposed method with other compression-based or information-theoretic methods used in different domains.

In addition, this work evaluates the performance of the proposed method on various molecular classification and regression tasks from the MoleculeNet benchmark, as well as a binding affinity prediction task using the PDBbind data set. However, the paper does not provide a detailed analysis of the limitations or failure cases of the proposed method or investigate the sensitivity of the method to different hyperparameters or settings. The paper also provides no ablation studies or qualitative analysis of the learned representations. In addition, no evaluation of the k parameter was explained for either classification or regression settings.

Some technical terms and symbols without proper definitions or explanations are utilized, such as TMAP, NCD, and Gzip. While the proposed method has some potential applications in chemical information retrieval and drug discovery, its impact on these fields is still limited by several factors. For example, the method relies on the availability and quality of SMILES strings or other string representations of molecules, which may not be accurate or complete for all molecules. The method also does not capture some important aspects of molecular structure and function, such as chirality, conformational flexibility, and electrostatic interactions.

**Questions:**

The paper could explore more advanced compression algorithms or information-theoretic methods designed explicitly for molecular data, such as arithmetic coding, Lempel-Ziv-Welch coding, or Kolmogorov complexity. The paper could also investigate integrating these methods with other machine learning models or domain-specific knowledge to improve their performance or interpretability.

The paper could conduct more extensive experiments on more diverse data sets that cover a more comprehensive range of molecular tasks and modalities. The paper could also perform a more detailed analysis of the learned representations using techniques such as clustering, visualization, or transfer learning. The paper could also compare the proposed method with other state-of-the-art methods that use different types of input or output.

The paper could provide more detailed definitions and explanations of technical terms and symbols used in the paper. The paper could also use more concrete examples and illustrations to clarify ambiguous or vague expressions. The paper could also provide more detailed descriptions of the experimental setup, hyperparameters, and evaluation metrics used in each task.

The paper could investigate how to address some of the limitations or challenges faced by the proposed method in real-world applications. For example, the paper could explore handling rare or unseen molecules using transfer or active learning. The paper could also investigate how to incorporate additional features or constraints into the compression-based method to capture more aspects of molecular structure and function. Finally, the should contextualize the importance of molecular similarity in drug discovery and how it is the best predictor of activity in most real word applications, see for example (https://doi.org/10.1021/jm401411z)

---

### Official Review · Reviewer_o9gf · 2023-11-06

**Soundness:** 1 poor
**Presentation:** 3 good
**Contribution:** 2 fair
**Rating:** 3
**Confidence:** 5

**Summary:**

(Jiang et al. 2022) proposed a Gzip based representation method for text classification tasks. Since chemical molecules can be represented as text (SMILES), this paper examines whether the same benefits translate into the chemical domain.

**Strengths:**

- The method is simple and not computationally intensive.
- The problem is well motivated and the writing is clear

**Weaknesses:**

First, the method is a straightforward adaptation of the method from Jiang. et all for Molecules.

The comparison with other methods are not thorough. The comparison is performed only against ChemBERTa-1, ChemBERTa-2 and Grover. These are not the state of the art methods for these tasks. While the authors cite state of the art methods like Molformer-XL, no comparison is done against these methods. And when compared to even simpler methods like SVM on molecular fingerprints, the performance of the proposed method is not better. For example on the BACE task, SVM has an AUROC of 86.2 whereas Molzip has an AUROC of 66.7 (Numbers are from the Molformer-XL paper). In fact, in the BACE task, all the methods compared in the Molformer-XL paper (are better than MolZip. In the BBBP task, the best performance of Molzip is 69.2 whereas all the baseline methods (again taken from the Molformer-XL paper) are better than Molzip (with the exception of Chemberta), with Molformer-XL achieving 93.7 AUROC.

Similar statements can be made about the regression task as well.

It is generally expected gzip will have some predictive power as it internally builds a statistical model on the textual features. This paper does not show that the performance of the method is much better than that expectation.

**Questions:**

For Tox-21, only one task (out of the 12 tasks) is considered.  Any reason for this?

As a general suggestion, I would advocate for a more thorough comparison with other methods. That will help the reader understand the performance of this method in relation to other methods. Right now, the paper presents a very misleading picture on performance.

---

### Meta-Review · Area_Chair_UQ2k · 2023-12-10

**Metareview:**

This paper extends the work of (Jiang et al. 2022) of text representation via gzip compression to molecules based on the smiles representation of molecules .

 Reviewer o9gf argued that the innovation is limited since its straightforward application of the Jiang et al, and claimed that the benchmarking in the paper is not thorough with respect to baselines  an state of the art such as MolFormer-XL, and that the results are not competitive with simple baselines such as the ones trained with SVM.

Reviewer 1REg , while positive on the work suggested comparisons with other compression baselines and state of the art methods and other datasets. They suggested  also more ablations studies on the hyper-parameters of the method to understand when it fails.

Reviewer 5hym echoed similar assessment in terms if 1) limited contributions 2) lack of baselining with respect to  state of the art and w.r.t to fingerprints with knn 3) lack of uncertainty quantification.

Reviewer ovMF challenged the authors regarding the framing of the paper on why compression method works and if it is an artifact of simplicity of tasks at hand. The reviewer proposed another distance based on the  compressed representation and suggested to have simpler baselines to understand the proposed method.

The paper presents an interesting angle on the use of compressed representation in molecular domain, reviewers suggested many improvements to the paper.  Authors did not submit a rebuttal. We hope the authors incorporate reviewers feedback and strengthen their submission and encourage them to submit the work to an upcoming venue.

**Justification For Why Not Higher Score:**

paper needs better framing, inclusion of more baselines  tasks and ablation studies. Also deepening the understanding on when these compressed representation are effective,

**Justification For Why Not Lower Score:**

N/A

---

### Decision · Program_Chairs · 2024-01-16

Reject